# Heterogeneous Communication Network Architecture for the Management of Electric Vehicle Charging Stations: Multi-Aggregator Management in Microgrids with High Photovoltaic Variability Based on Multiple Solar Radiation Sensors

**DOI:** 10.3390/s24123768

**Published:** 2024-06-10

**Authors:** Miguel Davila-Sacoto, Luis Hernández-Callejo, L. G. González, Óscar Duque-Perez, Ángel L. Zorita-Lamadrid, Danny Ochoa-Correa

**Affiliations:** 1Department of Agricultural and Forestry Engineering, Duques de Soria University Campus, University of Valladolid, 42004 Soria, Spain; miguelalberto.davila@estudiantes.uva.es; 2Department of Electrical, Electronics and Telecommunications Engineering, Campus Balzay, University of Cuenca, Cuenca 010107, Ecuador; luis.gonzalez@ucuenca.edu.ec (L.G.G.); danny.ochoac@ucuenca.edu.ec (D.O.-C.); 3Department of Electrical Engineering, School of Industrial Engineering, University of Valladolid, 47011 Valladolid, Spain; oscar.duque@uva.es (Ó.D.-P.); zorita@uva.es (Á.L.Z.-L.)

**Keywords:** electric vehicle aggregator, electric vehicle, photovoltaic generation, microgrid, EV charging station, Vehicle-to-Grid

## Abstract

Electric power systems with a high penetration of photovoltaic generation and a relevant fleet of electric vehicles face significant stability challenges, particularly in mountainous areas where the variability of photovoltaic resources is pronounced. This study presents a novel methodology to strategically place electric vehicle aggregators along a feeder. This approach considers electrical variables and the dynamics of cloud movements within the study area. This innovative methodology reduces the substation’s power load demand and significantly improves the end user’s voltage levels. The improvements in voltage regulation and reduced demand on the substation provide clear benefits, including increased system resilience, better integration of renewable energy sources, and enhanced overall efficiency of the electric grid. These advantages are particularly critical in regions with high levels of photovoltaic generation and are important in promoting sustainable electric vehicle charging infrastructure. When analyzing different load scenarios for the IEEE European Low Voltage Test Feeder system, the consideration of distributed aggregators based on cloud movements decreased the power required at the substation by 21.25%, and the voltage drop in loads was reduced from 6.9% to 4.29%. This research underscores the critical need to consider both the variability and geographical distribution of PV resources in the planning and operation of electrical systems with extensive PV generation.

## 1. Introduction

The growth in electric infrastructure integrating distributed renewable energy sources, particularly solar photovoltaic (PV) systems, has significantly increased. This trend introduces complex system monitoring, optimization, and analytical assessment challenges. Power grids that heavily rely on solar PV face unique challenges due to the unpredictable nature of solar irradiance. Such unpredictability can cause voltage instabilities at different grid nodes [1]. Simultaneously, the global trend towards adopting electric vehicles (EVs) adds further complexity to the grid. EVs not only impose a considerable demand on the electrical system, but also offer the potential to function as mobile energy storage units. They can provide vital ancillary services to the electrical grid, especially during emergencies, thereby enhancing the resilience and stability of the grid. The adequate integration of EVs into grids with significant solar PV penetration has been the focus of extensive research, aiming to mitigate the voltage fluctuations caused by the erratic behavior of solar radiation [2,3]. Achieving this integration requires formulating and implementing specific charging strategies, particularly crucial in isolated grids where renewable sources are a significant portion of the energy portfolio [4].

To address these integration challenges, the adoption of Vehicle-to-Grid technology (V2G) has been proposed [5]. In this innovative framework, EVs, primarily through the infrastructure of EV charging stations or electric vehicle supply equipment (EVSE), become dynamic agents capable of contributing to and drawing energy from the grid. This dual capability is essential for maintaining optimal voltage levels, ensuring frequency stability, and managing peak energy demands. To efficiently control the complex dynamics between EVs and the electrical grid, aggregator systems are employed [6]. These systems make strategic decisions about energy transfer between EVs and the grid. By analyzing various electrical and economic factors, including energy costs, they aim to optimize the energy flow from EVs in a way that is consistent with the operational requirements and limitations of the electrical grid.

Incorporating a wide range of telecommunications technologies is critical when developing a heterogeneous communication network architecture for managing EVSEs. These technologies include Wi-SUN, LoRa, Ethernet, fiber optics, and cellular networks like 3G and 4G. Each technology offers different coverage, bandwidth, latency, and dependability advantages, making them suitable for different roles within the EV charging domain.

Recent advancements in telecommunication technologies for microgrid systems have been extensively reviewed. For instance, in [7], the authors highlight the importance of robust communication infrastructure in maintaining the stability and efficiency of networked microgrids. This review highlights the need for secure and reliable communication channels to facilitate real-time data exchange and control operations in smart grid environments.

The exploration of LoRaWAN-based communication systems to facilitate interactions between aggregators and EVs within V2G frameworks is well documented. LoRaWAN is recognized for its cost-effective approach, characterized by a wide range and minimal energy expenditure [8], making it suitable for large-scale deployments in urban and rural settings. Additionally, Narrowband IoT (NB-IoT) has emerged as a power alternative for V2G communications. NB-IoT offers advantages regarding deep penetration and improved power efficiency, which are crucial for reliable communication in dense urban environments where building penetration is required. Both technologies have the potential to significantly reduce operational costs and improve data transmission efficiency in smart grid environments.

Recent studies have further demonstrated the performance and applicability of these technologies in different scenarios. For example, research conducted in [9] indicates the superior reliability of NB-IoT in mining environments near urban settings, while Ref. [10] highlights the cost-effectiveness of LoRaWAN in rural and suburban deployments. These studies provide a comprehensive overview of how diverse communication technologies can be optimized to meet the specific needs of electric vehicle charging management within V2G systems. Integrating these technologies into a heterogeneous network paves the way for establishing a resilient and flexible communication infrastructure.

Infrastructure based on heterogeneous communication networks is suited to address the multifaceted requirements of microgrid management [11]. Such a network improves the smooth incorporation of EVs into the electrical grid, fine-tuning the process of vehicle charging and discharging in response to the availability of renewable resources, grid status, and other dynamic factors. Recent studies have highlighted the significance of advanced communication infrastructure in optimizing V2G systems. For instance, Ref. [12] discusses the role of distributed control techniques in heterogeneous microgrids, emphasizing the importance of reliable communication protocols for enhancing system performance and resilience. Other studies focus on the implementation of hardware. For example, Ref. [13] explores the implementation of electronically steerable antennas in heterogeneous communication networks, demonstrating their potential to provide high-throughput, low-latency services, which are critical for microgrid applications. Additionally, Ref. [14] examines the communication requirements and protocols for secure and efficient operation in distributed microgrids, highlighting the ongoing advancements in smart grid communication technologies. Furthermore, Ref. [15] emphasizes the integration of networked microgrids linked through heterogeneous IoT communication networks, showcasing their impact on enhancing grid control and communication paradigms. These studies collectively highlight the importance of reliable and efficient communication infrastructures in supporting the seamless integration of EVs into the electrical grid, thereby optimizing energy management and enhancing grid resilience.

The research presented in this paper advances the understanding of how diverse communication technologies can cause delays in communication between an aggregator and an electric vehicle. Our study demonstrates the benefits of a multi-aggregator management model within microgrids experiencing significant solar PV penetration, addressing the communication system delays, the impact of cloud movements, and adjustments in power setpoints.

The methodology presented in this study for strategically placing EV aggregators offers several key advantages:Enhanced Voltage Regulation: Our methodology significantly improves voltage levels at user terminals by strategically placing EV aggregators. This is particularly important in microgrids with high PV penetration, where voltage fluctuations might be pronounced due to the intermittent nature of solar irradiance.Reduced Power Load Demand: The methodology effectively reduces the power load demand of the substation by optimizing the power flow between EVs and the grid. This relieves the load on substations and enhances the electrical grid’s overall stability and efficiency.Improved Integration of Renewable Energy Sources: Our approach facilitates better integration of renewable energy sources, especially solar PV systems, by mitigating the adverse effects of PV variability. This leads to a more resilient and reliable energy supply.Increased System Resilience: The innovative placement of EV aggregators helps to dampen fluctuations caused by intermittent solar resources, thereby increasing the resilience of the electrical system against unexpected changes in power generation.Optimized Energy Utilization: By employing multiple EV aggregators referencing diverse PV radiation signals, the methodology ensures optimal utilization of both EVs and PV systems. This dual optimization enhances energy efficiency and maximizes the benefits of distributed renewable energy sources.Sustainable EV Charging Infrastructure: The improvements in voltage regulation and reduced demand on substations are instrumental in promoting sustainable EV charging infrastructures, which are crucial for supporting the growing adoption of electric vehicles.Adaptability to Geographical Variability: The methodology considers the geographical distribution of PV resources, making it particularly effective in regions with significant topographical and meteorological variations, such as mountainous areas.

This article is structured as follows: Section 2 develops the problem definition of delays caused by heterogeneous networks and cloud movement in PV generation and electric vehicle charging. Section 3 details the methodology used to develop the simulation system, including the parameters employed. It covers the selection of software tools (like Python and OpenDSS), the design of the simulation environment, the criteria for choosing specific models and scenarios, and the assumptions made to facilitate the simulations. This section also explains how the simulation integrates electrical and communication systems to evaluate their interplay and impact on the microgrid’s performance. Section 4 outlines the case study used to apply and test the simulation system. This part of the paper describes the specific microgrid configuration, the geographical location, the characteristics of the PV integration, the types of EVs involved, and the setup of communication networks. It provides the context and details required to understand the practical application of the simulation system developed in Section 3. Section 5 delivers the findings of the study. It presents data, graphs, and analyses, illustrating the simulation response to different scenarios, including the effects of telecommunication delays and multiple photovoltaic sensors. This section also discusses the system’s reliability under different conditions, the efficiency of the V2G integration, and how cloud movements impact the system.

## 2. Problem Definition

For the development of this study, the following issues of a microgrid with high photovoltaic penetration and connected EVs were analyzed. The problem caused by the high variability of the PV resource in microgrids with high penetration of PV generation is discussed. This includes a review of the performance of two PV generators analyzed in this research, where the delay in generation caused by cloud movement is observed. Secondly, the problem with telecommunication delays in heterogeneous networks caused by different technologies is analyzed. Finally, the hardware delays caused by different EV types of hardware are described.

### 2.1. High Variability of the Photovoltaic Resource

Microgrids with high PV penetration, particularly in mountainous regions like Cuenca, Ecuador, where the experiment in this study took place, experience rapid voltage variations due to the high variability of the PV resource. This variability is further exacerbated in mountainous areas due to the unique geographical and meteorological conditions that affect solar radiation levels, with dense cloud cover notably contributing to the phenomenon. These microgrids require integrating battery storage systems or auxiliary grid services to maintain stable voltage levels for the end user. These systems help buffer the fluctuations caused by the intermittent solar resource, ensuring a reliable power supply and enhancing the overall resilience of the grid.

The results of this analysis revealed that, due to the frequent presence of cloud cover in the mountainous Andes environment, the power generated by the PV system experiences substantial oscillations. These oscillations were identified as variations that fall within a range oscillating between 54% and 72% of the nominal capacity of the installation. Additionally, when examining the direct current component of the generated power, it was observed that variations in the solar system occur at intervals on the order of 15 s [16], with peak values of 800 Wm^−2^/min and an average of 275 Wm^−2^/min (see Figure 1). The data correspond to the 35 kWp PV generator from the Microgrid Laboratory at the Balzay Campus of the University of Cuenca, Ecuador.

The variation of the PV resource must also consider the movement of clouds across the area covered by the microgrid. The delay effect caused by cloud movement was measured by comparing data from two PV generators equipped with weather stations and solar radiation sensors, identified as PV1 and PV2 in Figure 2. The generators are located in Cuenca, Ecuador, at coordinates 2°53′52.80” S and 79° 0′52.64” W. The two generators, PV1 and PV2, are spaced 5.6 km apart. PV1 belongs to the Microgrid Laboratory at the University of Cuenca and boasts a peak power of 35 kW. Conversely, PV2 is owned by the city’s electric company and operates at a peak power of 15 kW.

The satellite image of Cuenca reveals a cloud covering part of the city. Even on a partially clear day, clouds cast shadows over scattered areas of the city, highlighting the spatial variability of solar radiation. It is essential to emphasize that the presence of clouds can significantly impact solar energy generation, as the shadows cast by the clouds reduce the amount of solar radiation incident on the surfaces of PV panels, thereby affecting energy production. Now imagine the movement of the cloud from northeast to northwest at a speed *v*; the cloud will cover parts of the city in its path. However, during the day, there will be areas that are not covered by the clouds. In this scenario, the cloud will cover PV2 first and then, after a time *t*, it will cover PV1.

The measured power values generated by PV1 and PV2 are displayed in Figure 3. It is noted that there is a delay between the two generators, which can be directly attributed to the area’s cloudiness. The delay, in this case, averages 20 min.

### 2.2. Telecommunication Delays in Heterogeneous Networks

A significant challenge is the communication platform encompassing the interaction between the EV and the network control agents [17]. This requires a communication infrastructure that responds to the system controller’s signals to ensure energy quality. To achieve this, such systems employ various communication technologies. For instance, reference [18] uses the Industrial Internet as a communication channel between the distribution or the dispatch center and vehicles, and Ref. [8] uses a LoRaWAN radio frequency system. In contrast, Ref. [19] uses another approach incorporating a hybrid model where a 5G network acts as the primary connection point, facilitating wireless interactions between the vehicles and the power distribution company. The effectiveness of this hybrid model relies on the 5G network’s ability to transmit messages with minimal delay. Table 1 compares the most used communication systems in VGI (Vehicle Grid Integration).

Another crucial factor is the communication delay between the aggregator or the central control office of the DSO (Distribution System Operator) and the charging stations. Figure 4 illustrates two commonly used communication infrastructure options for these types of networks. It shows the delays between different pieces of equipment and phases of the network, where it is expected to have a radiofrequency WAN (Wide Area Network) for last-mile communications. In a second instance, two communication options are presented: one that uses a network from a standard internet provider, typically that of the user or the same distributor, which communicates directly with the EVSE for charging, and another option that utilizes the FAN (Field Area Network) structure, leveraging the smart metering network (AMI—Advanced Metering Infrastructure) for EVSE communication (see Figure 5). Delay times are calculated using Equation (1).
(1)$$d=\sum_{i=1}^{i}d_{i}$$

### 2.3. Hardware Delays

In recent years, there has been growing interest in using EVs to offset fluctuations in the electricity grid, necessitating understanding EVs’ responsiveness to changes in charging setpoints. This is used to leverage the energy stored in their batteries to support the grid. Consequently, studies like [20] have delved into EVs’ dynamic response in an isolated power system when engaged in frequency control. This study adopts a V2G approach for employing EV charging stations in both primary and secondary frequency control in a single bus system with a 100 kW synchronous generator, a transformer, and a static load, where EVs are connected for frequency compensation. The findings indicate that EVs can adequately respond to Primary Frequency Control (PFC) and Load Frequency Control (LFC), enhancing the maximum frequency reduction during disturbances with durations shorter than 3 s. A similar investigation in [21] addresses inter-area frequency variations, noting improvements in frequency stabilization times of up to 50%.

In voltage or power control applications, adjustments to the vehicle’s charging setpoint need to occur rapidly. This aspect was investigated by examining the response times of two EVs, a KIA Soul EV and a BYD E5 400. Tests were conducted to determine their reaction times to changes in charging setpoints. These tests showed varied dynamic responses among the vehicles; for instance, one EV responded to a load setpoint change from 50% to 30% in approximately 2 s, whereas the other achieved a response time of 0.68 s, as depicted in Figure 6. Additionally, the charging process initiated at 5.5 kW. Then it transitioned to 2.8 kW (slow charging mode), focusing the measurement on the time it takes for the vehicle’s charge controller to adjust to the new setpoint.

## 3. Methodology

The integration of EVs into microgrids with high PV penetration has not been extensively covered in the literature. Furthermore, the effects of minor delays caused by heterogeneous network communications, hardware delays due to power setpoint changes in EVs, and the impact of cloud movements on networks that utilize a single PV sensor for power setting control, which is crucial in Vehicle-to-Grid (V2G) applications, have not been adequately considered. To address these scenarios, we developed a comprehensive simulation platform using Python and OpenDSS. This platform enabled detailed modeling of a microgrid’s electrical and communication systems, facilitating the analysis of how these systems interact and impact overall performance. The simulation was based on the following points:Development of a simulation platform: A simulation platform was created using Python and OpenDSS to simulate the electrical system while considering telecommunication delays. This platform allows for the detailed modeling of the electrical and communication systems in a microgrid, enabling the analysis of how communication delays and hardware responsiveness impact the integration of EVs and photovoltaic systems.Analysis of multiple PV sensors’ effects: The study also examined the impact of using multiple PV sensors within the network to meet V2G directives. This approach helps understand how the spatial distribution of sensors can mitigate the effects of cloud movements and other environmental factors on the accuracy and efficiency of PV power generation and its integration into the grid.

By implementing this methodology, the research aims to fill gaps in the existing literature by providing new insights into the complexities of integrating EVs into PV-enriched microgrids, particularly considering the dynamic and sometimes unpredictable nature of renewable energy sources and the communication networks that support them. This could lead to more robust and responsive microgrid systems, optimizing energy production from renewables and the beneficial role of EVs within the grid.

A co-simulation environment combining OpenDSS and Python served as the simulation platform. This tool is under continuous development by the Universidad de Valladolid and Universidad de Cuenca, and it is accessible for download at https://github.com/davilamds/EVPVSimulation (accessed on 6 June 2024). The platform considers the impact of integrating both PV systems and EVs, setting parameters for system activation (for preliminary analysis), and managing the communication and functionality of each component (refer to Figure 7). The interaction between Python and OpenDSS was developed considering:Data initialization and input: Python was used to initialize the simulation parameters, including the microgrid configuration, the placement of EV aggregators, and the distribution of PV systems. These parameters were fed into OpenDSS to set up the electrical network model.Simulation control: Python scripts controlled the simulation process in OpenDSS. This included starting and stopping the simulation, adjusting time steps, and modifying system parameters in real-time based on predefined scenarios and dynamic events such as cloud movements and EV charging demands.Data exchange and processing: During the simulation, OpenDSS generated detailed electrical data, including voltage levels, power flows, and load demands at various nodes. Python retrieved this data through COM interfaces, processed it, and analyzed it to assess system performance. Python’s data handling capabilities allowed us to evaluate the effects of telecommunication delays and hardware responsiveness.Integration of multiple PV sensors: Python managed the integration of multiple PV sensors by simulating their readings and incorporating them into the control with OpenDSS. These readings were used to adjust the power setpoints for EV charging and discharging by OpenDSS, ensuring that the system responded dynamically to changes in solar irradiance.Evaluation of communication and hardware delays: The platform accounted for various delay characteristics. Python scripts introduced these delays into the simulation, enabling the analysis of their impact on system performance. OpenDSS processed the delayed signals to reflect the real-time effects on voltage regulation and power flow.Simulation of EV aggregator control: Python scripts implemented the control logic for EV aggregators, passing the activation and deactivation of EVs to OpenDSS.

In order to ascertain the timing of EV connection events, this study employed a bi-modal probability curve. This approach enables the automatic generation of connection probabilities, highlighting two distinct peaks that reflect the typical connection patterns of EV users (refer to Figure 8). Furthermore, the study considered changes in the delay time associated with EV charging power setpoint commands. This consideration is crucial for accurately modeling and predicting the behavior of EVs within the electrical grid, ensuring that simulations closely mirror real-world scenarios.

Regarding communication parameters, the platform accounts for the delay between the aggregator and the EV charging station. It calculates transmission and propagation delays while also considering queuing and processing delays (refer to Figure 9). These delays are primarily determined by the characteristics of the transmission medium and the number of communication nodes present. This consideration becomes critical in grids with a high penetration of EVs, as requests to alter the charging setpoint can result in significant processing and queue times.

## 4. Case Study

A strategy identified as an effective means to mitigate the fluctuations in voltage and power levels that arise from the variability of PV solar resources involves leveraging EV aggregator systems. These systems coordinate the implementation of the Vehicle-to-Grid (V2G) concept to rectify the variations observed within the system [22]. Within the scope of this study, it was hypothesized that such an aggregator should be able to account for changes in the availability of PV solar resources attributed to cloud movements over a specific area.

For this analysis, three geographical areas were designated as Z1, Z2, and Z3, where a pattern of solar radiation movement across a three-hour timeframe was monitored. The specifics of these zones and the dynamics of solar radiation movement are illustrated in Figure 10. Integrating this dynamic into the design and functionality of the EV aggregator introduces a novel strategy for mitigating the variability in PV generation and enhancing the electrical system’s adaptability to changes driven by meteorological conditions. This research enriches our comprehension of managing spatial and temporal variations in solar radiation within energy systems that significantly incorporate PV solar energy.

Moreover, concerning the control implemented by the aggregator, the following scenarios were examined:A singular aggregator that manages all EVs on a feeder, referencing a single PV resource signal.Several aggregators each manage segments of EVs on a feeder, all referencing a single PV resource signal.Several aggregators each manage segments of EVs on a feeder, each using multiple PV resource signals as references.

This analysis took place using the IEEE European Low Voltage Test Feeder system, configured with 14 EVs, each with a power capacity of 7.4 kW, and 14 PV systems, each with a capacity of 1 kVA. These specific power values were selected to demonstrate the effects discussed in this study while preventing excessive distortion of the analyzed IEEE case, given its nature as an underground system characterized by very low loads. EVs were simulated as batteries, and EV aggregators were simulated as storage controllers. For the PV systems, the default simulator model was employed (refer to Figure 11). The EVs were scheduled to connect from 13:00 to 17:00, mirroring a typical connection pattern observed in public parking areas during their second phase of connection to the system. This connection timeframe was also deliberately aligned with the fluctuations in PV resource availability.

## 5. Results

The simulation conducted within the OpenDSS framework spanned a 24 h timeframe, incorporating the electrical loads specific to the IEEE system. The initial analysis focused on the power output from the substation serving the circuit under examination, as depicted in Figure 12. This analysis noted that both scenario 1 and scenario 2 achieved their peak power output at 16.63 h, recording a power level of 115.8 kW. On the other hand, scenario 3 demonstrated a reduced power demand, peaking at 16.67 h with a power output of 91.19 kW. These findings show that the approach involving multiple aggregators and considering diverse PV solar radiation signals effectively lowers the power demand on the substation and enhances energy utilization from both EVs and PV systems.

This outcome underscores the significance of sophisticated energy management strategies that leverage renewable energy sources alongside the capability of EVs to serve as both energy storage and demand response mechanisms. Adopting strategic coordination measures, as seen in scenario 3, facilitates a more efficient electrical system operation. This solution alleviates the load on the substation and maximizes the use of renewable resources. The results affirm the effectiveness and potential of EV aggregators in improving the quality and stability of electrical power systems that incorporate a significant amount of PV solar energy.

Examining the voltage at load 349 (refer to Figure 13) reveals that scenario 3 results in a lesser voltage drop than the other scenarios, registering a voltage of 233.9 V. This pattern remains consistent when EVs initiate charging or when vehicles are providing energy back to the system. Specifically, the voltage drop in scenario 3 is improved by approximately 2.6% compared to that in scenario 1 and 2.2% compared to that in scenario 2. The average voltage at critical load points was maintained closer to the nominal voltage level, highlighting the superior performance in voltage regulation.

Additionally, EVs’ state of charge (SOC) and charging patterns were examined. It was noted that there is a minimal variation in SOC, fluctuating between 33.55% to 33.42% (refer to Figure 14). However, upon analyzing the duration EVs remain in charging and discharging states, it becomes apparent (see Figure 15) that vehicles commence charging and discharging at varying times.

Upon examining the specific charging and discharging durations, Table 2 indicates that charging periods remain consistent across all scenarios. Nonetheless, the discharging interval is reduced in scenario 3, implying diminished battery wear for the EV. Additionally, Table 2 outlines the outcomes related to substation power and the minimum load voltage. It is evident from the data that, in scenario 3, the substation expends less power to meet the system’s load requirements and enhances the minimum system voltage when compared to the baseline scenario.

The experiments focused primarily on capturing bus voltages across various delays by employing the co-simulation platform used for this study. Given the variance in delay times among different EV manufacturers and communication tiers [23], a simulation incorporating diverse delay time values was performed (refer to Figure 16). It was noted that with extended delay times in updating the power consumed by the EV, the voltages at the customer buses tend to rise. In contrast, with prolonged delay times, lower voltages are observed.

Summarizing, the following quantitative results were obtained:Voltage Drop Improvement: Scenario 3 showed a 2.6% reduction in voltage drop compared to scenario 1 and a 2.2% reduction compared to scenario 2.Minimum Voltage Levels: Scenario 3 maintained a minimum voltage level of 233.9 V, while scenarios 1 and 2 recorded lower minimum voltage levels of 227.3 V.Power Demand Reduction: Scenario 3 reduced the power demand on the substation to 91.19 kW, compared to 115.8 kW in scenarios 1 and 2.

## 6. Discussion

The strategic placement of EV aggregators along a power feeder is instrumental in managing the variability of PV generation within microgrids. This innovative approach could leverage several mechanisms to enhance grid stability and optimize the use of renewable energy resources.

Dynamic Load Balancing: EV aggregators dynamically adjust connected EVs’ charging and discharging schedules based on real-time PV generation patterns. This balancing act helps to synchronize the energy supply from PV systems with the demand from EVs, ensuring that excess PV energy is stored during periods of high generation and released during low generation periods.Voltage Regulation: By responding to voltage fluctuations caused by intermittent PV generation, EV aggregators help maintain stable voltage levels across the microgrid. When PV output drops due to cloud cover or other factors, EVs can discharge energy to support the grid, mitigating potential voltage drops and enhancing power quality.Frequency Stability: EVs’ bidirectional energy flow capabilities, managed by aggregators, provide crucial ancillary services such as frequency regulation. This is especially important in microgrids with high PV penetration, where sudden changes in generation can impact grid frequency. Aggregators modulate EV charging and discharging rates to stabilize frequency fluctuations, ensuring consistent and reliable grid operation.Optimal Utilization of Renewable Energy: Coordinating EV charging times with periods of high PV generation ensures maximum utilization of renewable energy. This reduces dependency on non-renewable energy sources and improves the overall efficiency and sustainability of the microgrid.Mitigation of Peak Load: EV aggregators contribute to flattening the demand curve by shifting loads to off-peak times. This load management is beneficial when PV generation does not align with peak demand periods. By discharging EVs during peak demand, aggregators alleviate stress on the grid and optimize energy distribution.Enhanced System Resilience: The distributed energy storage capability provided by strategically placed EV aggregators enhances the resilience of the microgrid. This distributed storage ensures a reliable energy supply during low PV output or high demand periods, contributing to a more robust and stable microgrid infrastructure.

The strategic placement of multiple solar radiation sensors across a microgrid offers several key benefits in managing EV charging stations in areas with high PV variability:Improved Accuracy of Solar Radiation Data: Deploying multiple sensors provides high-resolution data on solar irradiance levels across different parts of the microgrid. These detailed data enhance the accuracy of PV generation predictions, allowing for more precise matching of EV charging schedules with periods of high solar output.Enhanced Energy Management: Detailed and localized solar radiation data enable the energy management system to synchronize EV charging with periods of peak PV generation. This synchronization maximizes the utilization of renewable energy and minimizes reliance on non-renewable sources, improving the overall efficiency of the microgrid.Better Load Forecasting: Multiple sensors contribute to more accurate forecasts of PV generation by capturing the spatial variability of solar radiation. Improved forecasting allows for proactive adjustments in EV charging and discharging schedules based on anticipated changes in PV output due to weather conditions.Increased System Resilience: The redundancy provided by multiple sensors enhances the resilience of the microgrid. In the event of a sensor failure or data inaccuracies from one sensor, the system can rely on data from other sensors to continue making informed decisions, ensuring continuous and reliable operation.Optimized Voltage Regulation: Real-time solar radiation data from multiple sensors support better voltage regulation within the microgrid. The system maintains stable voltage levels by dynamically adjusting EV charging and discharging based on precise irradiance levels, preventing voltage sags and surges.Enhanced Decision-Making for Aggregators: Aggregators use detailed solar radiation data to make informed decisions about the optimal times for EV charging and discharging. This results in improved efficiency and performance of the EV charging stations, ensuring that energy is used optimally and sustainably.

## 7. Conclusions

This study delves into the impact of PV resource variability and its geographical spread, which causes delays in peaks or troughs within the radiation curve, leading to fluctuations in power and voltage in electrical systems with significant PV generation penetration. Adopting multiple EV aggregators that account for PV radiation curves across different system regions was suggested to tackle this challenge. This approach shifts the timing of when EVs supply and draw energy, showcasing its ability to lower the demand on substations and improve the system’s voltage response.

In essence, this research underscores the critical need to consider both the variability and geographical distribution of PV resources in the planning and operating of electrical systems with extensive PV generation. The strategic use of multiple EV aggregators emerges as a key solution, offering substantial benefits in managing these systems, thereby enhancing their overall functionality and stability.

The effect of telecommunications delays caused by the network’s heterogeneity is considered, where it is observed that different total delays result in different voltage values at the user’s terminals. Therefore, this effect must be considered when placing radiation sensors and telecommunications equipment if the aim is for EV auxiliary services to help reduce voltage variations in networks with high PV penetration. The effect of hardware delays is also important, as it was observed that they differ between different manufacturers. So, for system frequency ancillary services, these delays have to be considered, and the aggregator should be able to know all the delays in the system. This study considered the hardware delay, but the specific hardware used in particular EVSEs or EVs was out of scope.

Looking ahead, acquiring more data from various weather stations, synchronized via GPS, is essential to extend this observation to other urban areas. Moreover, developing strategies for station placement based on the noted delays and executing broader simulations within the electrical system are imperative. Determining the optimal number of EV aggregators needed to improve the system’s response is another crucial step. Additionally, conducting a capacity hosting analysis will be vital to link the integration of EVs, PV generation, and the number of aggregators overseeing the system. This examination will aid in identifying the most effective setup for controlling EV and PV resources within the electrical grid.

The scalability of the approach used in this study is a critical aspect that needs to be addressed for practical deployment in real-world applications. For example, the simulation platform, which integrates Python and OpenDSS, is designed to handle complex calculations efficiently. The platform’s modular architecture allows for scaling up the number of EVs without significant degradation in performance. By optimizing data handling and computational tasks, the platform can simulate larger systems with hundreds of EVs while maintaining accuracy.

The scalability of the communication network is crucial when increasing the number of EVs. Technologies like LoRaWAN, NB-IoT, and 5G offer high scalability due to their extensive coverage and high data throughput capabilities. These technologies can support large-scale deployments by efficiently managing the increased data traffic from numerous EVs and PV sensors. The multi-aggregator approach is inherently scalable. Each aggregator can manage a subset of EVs, distributing the computational and communication load. This decentralized management ensures the system remains responsive and resilient even as the number of connected EVs increases. Aggregators can coordinate through hierarchical or peer-to-peer communication models to optimize overall system performance.

For practical deployment, the methodology must consider factors such as grid infrastructure, existing communication networks, and the density of EVs. Pilot projects and phased rollouts can help fine-tune the system for larger-scale implementations. Continuous monitoring and adaptive management strategies are essential for handling the dynamic nature of large EV fleets.

## 8. Future Work

Future research should focus on expanding the analysis of different communication protocols to determine the most effective ones for managing EV and PV integration. Additionally, increasing the sample size of electric vehicle brands will provide a deeper understanding of hardware delay effects. Further studies should explore advanced communication technologies, such as 6G, and investigate their potential benefits. Implementing AI and machine learning for predictive analysis and decision-making can optimize energy management. Cybersecurity measures need to be developed to protect against potential threats. Real-world pilot projects are essential for validating the proposed methodologies and identifying any practical challenges of using multiple sensor arrays.

## Figures and Tables

**Figure 1 sensors-24-03768-f001:**
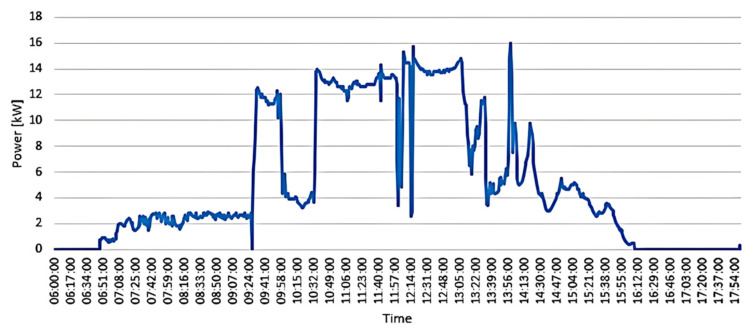
Photovoltaic power production at the microgrid laboratory.

**Figure 2 sensors-24-03768-f002:**
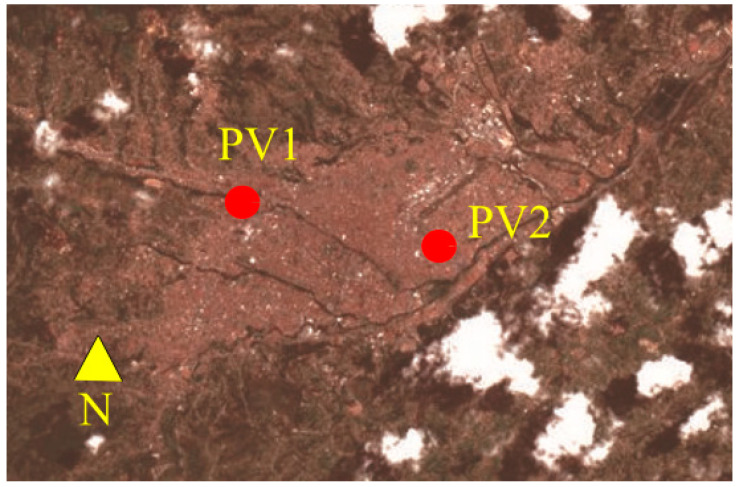
Location of photovoltaic systems. Source: Google Earth.

**Figure 3 sensors-24-03768-f003:**
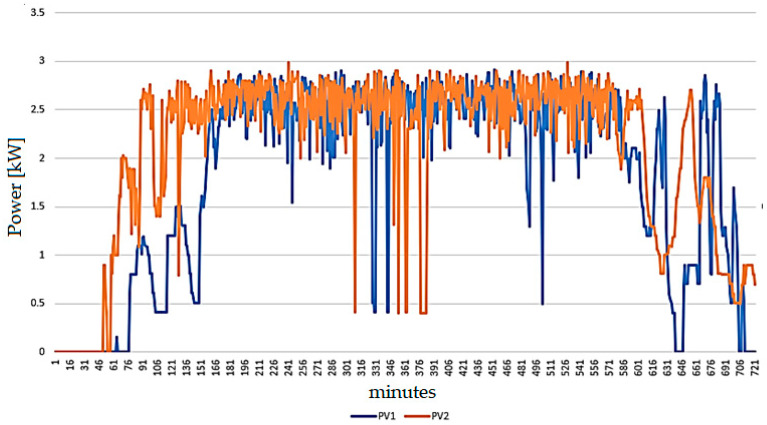
Daily production profile of the two photovoltaic installations under study.

**Figure 4 sensors-24-03768-f004:**
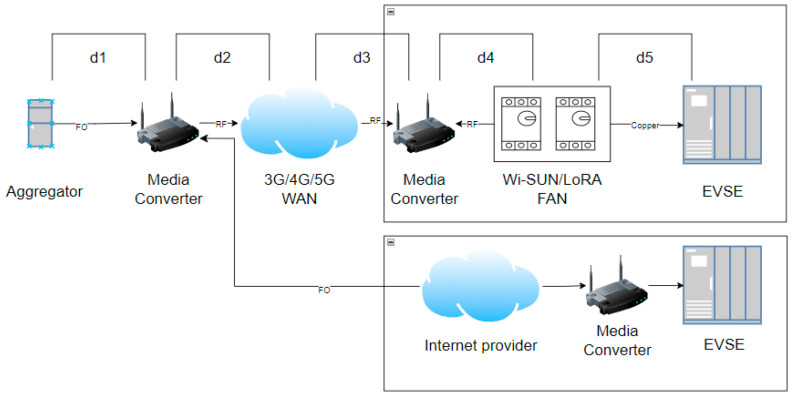
Communication delays.

**Figure 5 sensors-24-03768-f005:**
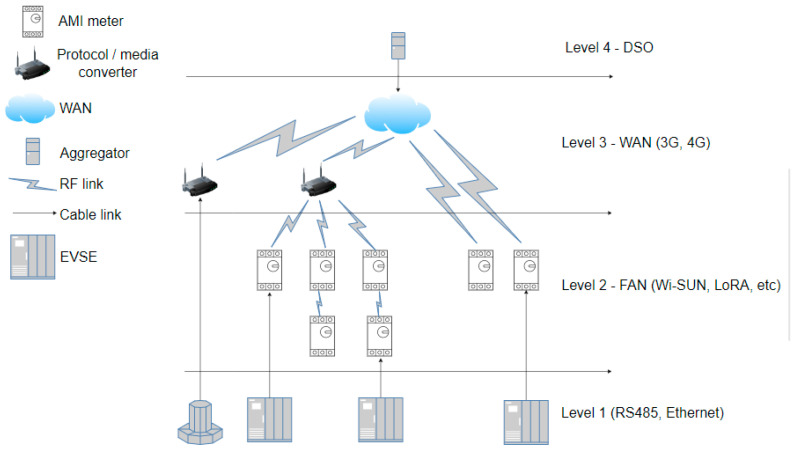
Communication tiers.

**Figure 6 sensors-24-03768-f006:**
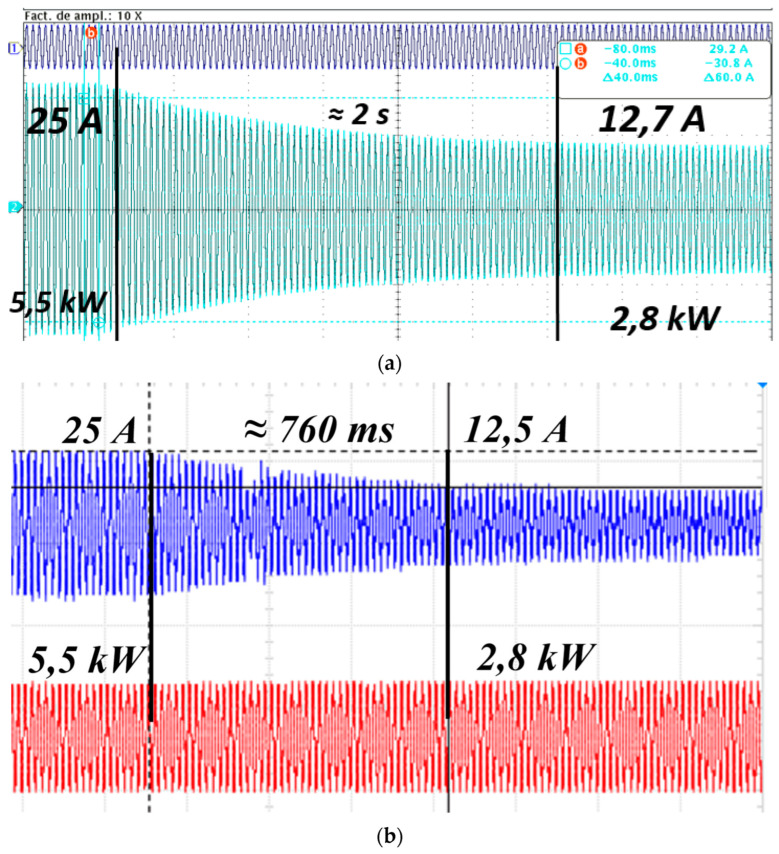
Dynamic behavior of EV charging when power setpoint changes: (**a**) EV No.1, (**b**) EV No. 2.

**Figure 7 sensors-24-03768-f007:**
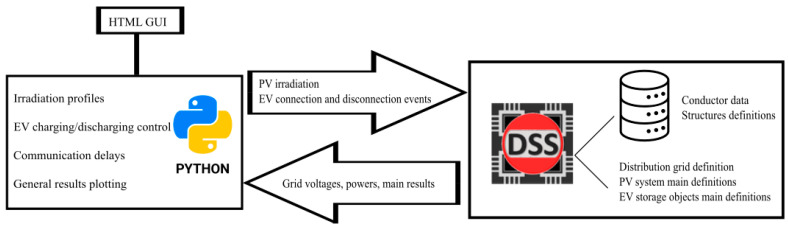
Integrated simulation platform design using Python and OpenDSS.

**Figure 8 sensors-24-03768-f008:**
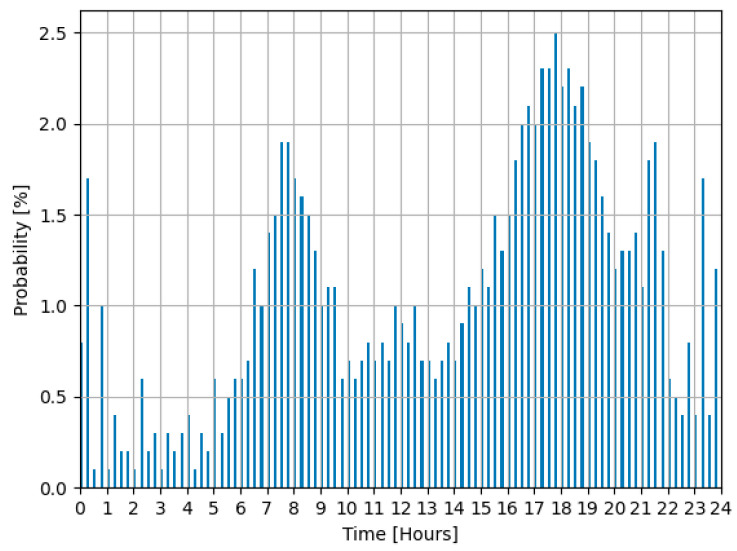
Electric vehicles’ connection probability according to [17].

**Figure 9 sensors-24-03768-f009:**
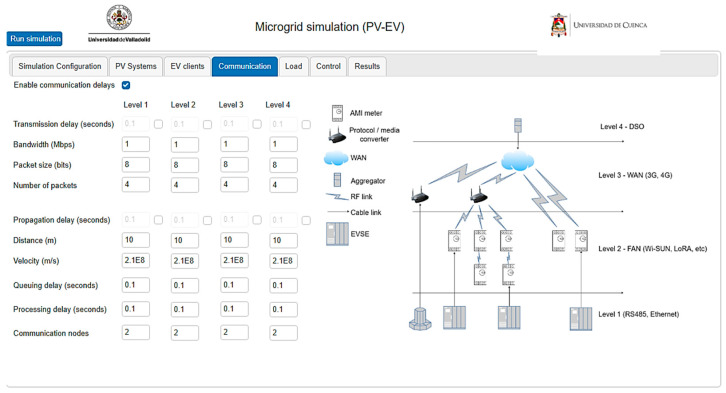
User parameters for communication delays.

**Figure 10 sensors-24-03768-f010:**
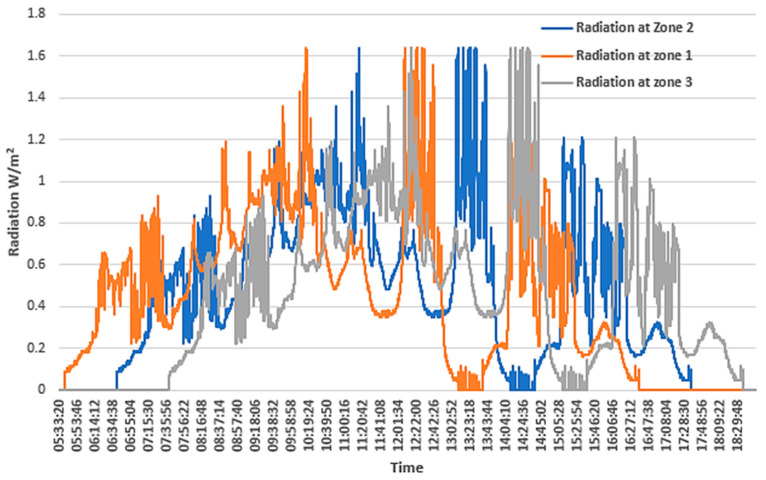
Solar radiation due to cloud movement.

**Figure 11 sensors-24-03768-f011:**
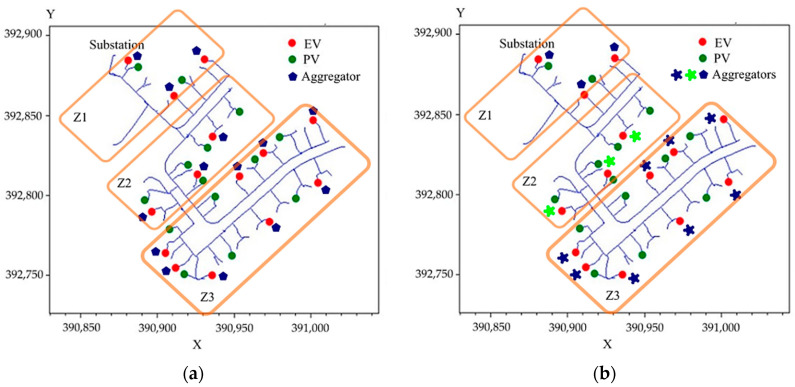
Simulated system with electric vehicle aggregators and cloud movement: (**a**) Case 1, (**b**) Cases 2 and 3.

**Figure 12 sensors-24-03768-f012:**
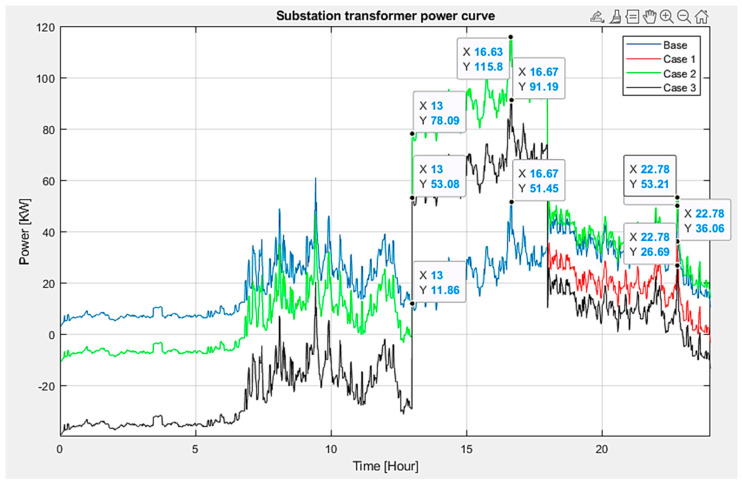
Power delivered by the substation.

**Figure 13 sensors-24-03768-f013:**
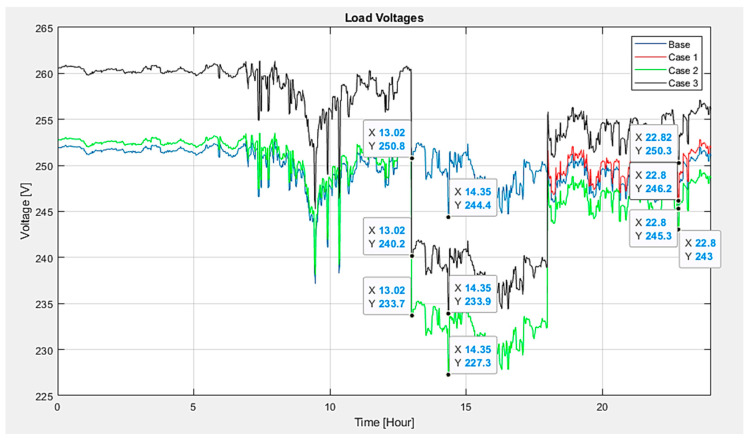
Voltage at system loads.

**Figure 14 sensors-24-03768-f014:**
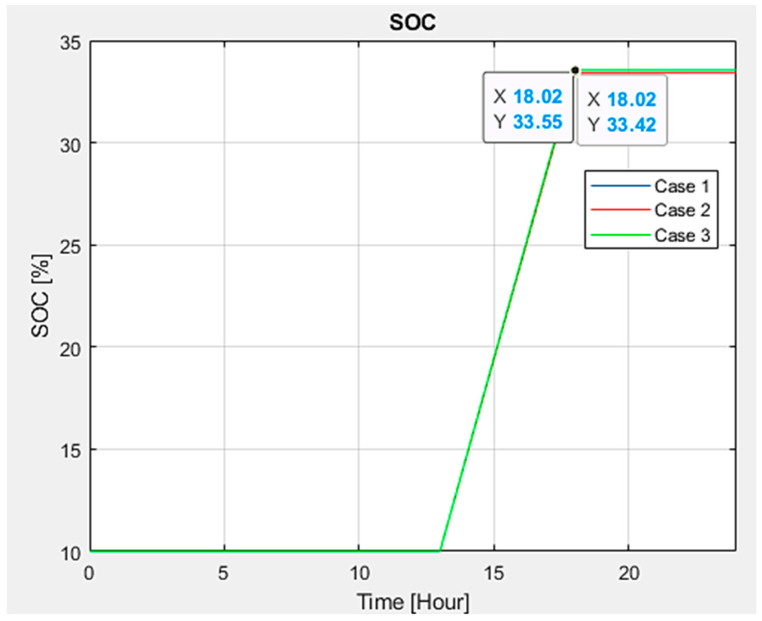
Electric vehicle state of charge.

**Figure 15 sensors-24-03768-f015:**
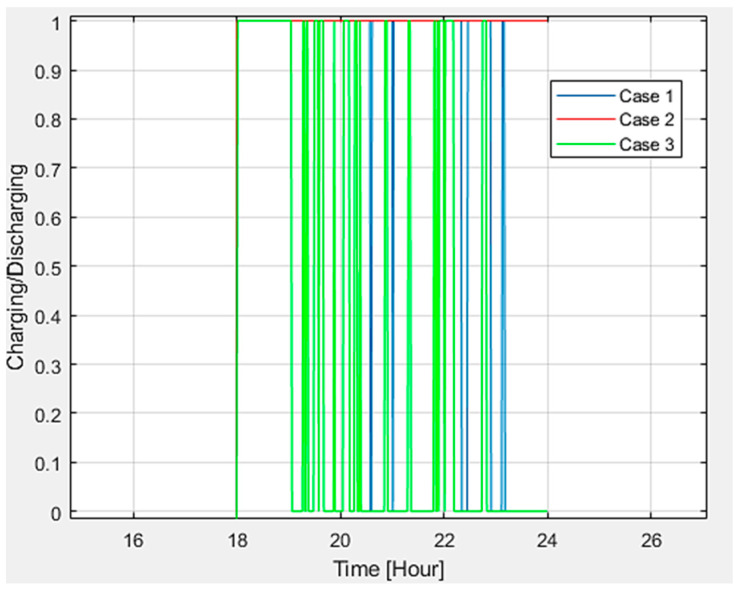
Charging and discharging events.

**Figure 16 sensors-24-03768-f016:**
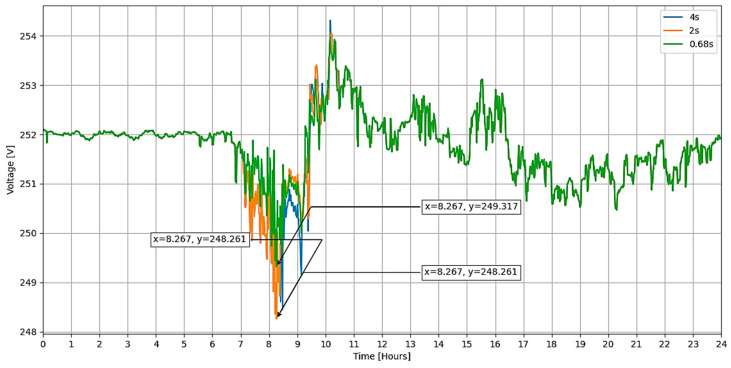
Voltage in the “load 349” connection bar for different EV charging setpoint delay times.

**Table 1 sensors-24-03768-t001:** Characteristics of communication systems used in VGI.

Communication System	LoRaWAN	Zigbee	4G	5G	Industrial Ethernet	FO
**Type**	Wireless	Wireless	Wireless	Wireless	Cable	Cable
**Data rate**	50 kbps	250 kbps	1 Mbps	100 Mbps	100 Mbps	10 Gbps
**Sensitivity**	−142 dBm	−126 dBm	−117 dBm	−113 dBm	−31 dBm	−19 dBm
**Range**	3 km	300 m	2.5 km	45 km	100 m	100 km (single mode)
**Average response time**	22.95 ms	10 ms	50 ms	5 ms	100 ms	0.1 ms per 10 km

**Table 2 sensors-24-03768-t002:** Charge and discharge time of different scenarios.

Scenario	Charge Time [minutes]	Discharge Time [minutes]	Substation Power [kW]	Minimum Load Voltage [V]
**Base**	-	-	51.45	244.4
**1**	300	288	115.8	227.3
**2**	300	361	115.8	227.3
**3**	300	119	91.19	233.9

## Data Availability

Data are accessible for download at https://github.com/davilamds/EVPVSimulation (accessed on 6 June 2024).

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
