# Peer review of "Heterogeneous Communication Network Architecture for the Management of Electric Vehicle Charging Stations: Multi-Aggregator Management in Microgrids with High Photovoltaic Variability Based on Multiple Solar Radiation Sensors"

_sensors, 2024, doi:10.3390/s24123768_

Round 1
Reviewer 1 Report
Comments and Suggestions for Authors
The abstract need more work, it would be better to enhance the abstract by mentionning the outcomes of thus study.
the author states '' innovative methodology'', please explain its advantages.
The manuscript claimed superior performance in voltage regulation. Can the authors quantify this improvement? Highlighting the specific metrics used to assess performance would strengthen this claim.
please, expand more in the technologies of telecommunication, refers to recent papers (A Comprehensive Review of Architecture, Communication, and Cybersecurity in Networked Microgrid Systems).
the graphs need extensive work, the axis not clear.
please update figure 6 either (a) and (b).
expand and explain more about the methodology used, and the ineraction between Python and DSS.
Figure 10 not clear, add a legend
the results need more clarification especially, figure 12
it ould be beneficial if you discuss the scalabilite of this methodology, (more than 10 EV connected)
Comments on the Quality of English Language
The english language need more work
Author Response
Answers in the attached document

Reviewer 2 Report
Comments and Suggestions for Authors
The idea for the paper is good, just some small suggestions:
- Please add more references for previous or related studies.
- Improve the picture quality as some of them are not readable or blurry on normal readability level.
- Please add more explanation to the case study and results.
- The conclusion need more detailed discussion and main takes from the research.
Author Response
Answers in the attached document

Reviewer 3 Report
Comments and Suggestions for Authors
Thank you for submitting your paper to Sensors. Here are a few suggestions to further increase the quality of the paper.
1)How does the placement of electric vehicle aggregators help manage the variability of photovoltaic generation in microgrids?
2)What are the key benefits of using multiple solar radiation sensors in managing electric vehicle charging stations in high photovoltaic variability areas?
3)How does the communication network architecture play a crucial role in the management of electric vehicle charging stations in microgrids with high photovoltaic variability?
4) Does the proposed methodology consider the actual charging station hardware characteristic in terms of increasing the system resilience and overall grid efficiency? For example, the magnetic component characteristic shown in [R1] and the charging infrastructure circuit topology shown in [R2].
[R1] Z. Luo, X. Li, C. Jiang, Z. Li, and T. Long, "Permeability-Adjustable Nanocrystalline Flake Ribbon in Customized High-Frequency Magnetic Components, " IEEE Trans. Power. Electron.,
[R2] H. Tu, H. Feng, S. Srdic, et al., "Extreme Fast Charging of Electric Vehicles: A Technology Overview," IEEE Transactions on Transportation Electrification, 2019, 5: 861-878.
Comments on the Quality of English Language
Please have another proofreading to ensure the correctness of the context.
Author Response
Answers in the attached document

Round 2
Reviewer 1 Report
Comments and Suggestions for Authors
The authors adreses most of the comments , please figure 10 need more ettention and try to add a legend to identify the curves.
Comments on the Quality of English Language
more improvments need for the English Language
Author Response
Thank you. Answer in the attached document

Reviewer 2 Report
Comments and Suggestions for Authors
The authors have updated the manuscript accordingly!
Author Response

(The authors gave the same response as above.)
